# Effects of Urban Forest Therapy Program on Depression Patients

**DOI:** 10.3390/ijerph20010507

**Published:** 2022-12-28

**Authors:** Poung-Sik Yeon, In-Ok Kim, Si-Nae Kang, Nee-Eun Lee, Ga-Yeon Kim, Gyeong-Min Min, Chung-Yeub Chung, Jung-Sok Lee, Jin-Gun Kim, Won-Sop Shin

**Affiliations:** 1Department of Forest Sciences, Chungbuk National University, Cheongju 28644, Republic of Korea; 2Graduated Department of Forest Therapy, Chungbuk National University, Cheongju 28644, Republic of Korea; 3Gwanghwamun Forest Psychiatric Clinic, Seoul 03156, Republic of Korea; 4Korea Forest Therapy Forum Incorporated Association, Cheongju 28644, Republic of Korea

**Keywords:** forest therapy program, urban forest, depression

## Abstract

Depression is a common serious mental health condition that can have negative personal and social consequences, and managing it is critical for treating depression patients. Forest therapy is emerging as a promising non-pharmacological intervention to improve mental health. However, although the effectiveness of forest therapy programs using forests far from the city has been proven, it is not well known that urban forests can be easily accessed in daily life. Therefore, this study aimed to examine the effects of an urban forest therapy program on depression symptoms, sleep quality, and somatization symptoms of depression patients. To evaluate this, a randomized controlled trial (RCT) design was employed. A total of 47 depression patients participated in this study (22 in the urban forest therapy program group and 25 in the control group). The Beck Depression Inventory (BDI), the Hamilton Rating Scale for Depression (HRSD), the Pittsburgh Sleep Quality Index (PSQI), and the Patient Health Questionnaire-15 (PHQ-15) were administered to each participant to assess the effects of the urban forest therapy program. The results of this study revealed that depression patients in the urban forest therapy program had significantly alleviated depression symptoms and improved sleep quality and somatization symptoms compared to the control group. In conclusion, this study demonstrates the possibility that the urban forest therapy program could be used as an effective non-pharmacological treatment to alleviate depression disorder.

## 1. Introduction

Depression is one of the most common mental health conditions and is assumed to be a major cause of disease burden by 2030 [1,2]. According to the World Health Organization [3], depression is a common disease worldwide, with 280 million people suffering from depression. Depression affects 3.5% of the world’s population, including 5.0% of adults and 5.7% of seniors aged 60 or older [4].

Common symptoms of depression are sadness, anxiety, insomnia, loss of energy, and lack of interest in life, which can adversely affect health [5]. In particular, people with depression are generally known to combine sleep disturbance with somatization [6,7,8]. In the case of sleep disorders, 90% of patients with depression will complain about sleep quality. About two-thirds of the patients experiencing major depressive symptoms experience insomnia, and about 40% of the patients are known to complain of sleep onset difficulties, frequent awakenings, or delayed or terminal insomniac problems [9,10]. Moreover, it turns out that about 15% of patients suffer from all three.

Somatization is the expression of mental phenomena as physical symptoms [11]. Disorders characterized by somatization continue to expand from symptoms that develop unconsciously and unconsciously to symptoms that develop consciously and voluntarily [12]. In the case of somatization symptoms, about two-thirds of patients with depression are also dominated by physical symptoms such as lack of energy and general pain, which are often caused by somatization [13,14]. Depression, sleep disorders, and somatization symptoms are interrelated. For example, Shim et al. [15] investigated the effects of somatization symptoms and sleep quality on depression and suicide risk indicators in 1937 Korean soldiers. There was a correlation between the degree of somatization symptoms, sleep quality, and depression, suggesting that somatization symptoms and sleep quality are potential indicators of depression. Therefore, considering that depression is a combination of various symptoms [5,16], interventions designed to prevent or treat depression should not focus on just one symptom.

Common treatments for depression are mainly medication and psychotherapy [17]. In particular, antidepressants play an important role in the treatment of depression but have disadvantages in use, such as cardiovascular disease, metabolic abnormalities, sexual deterioration [18,19,20], concerns about antidepressant drug resistance, and high recurrence rates [21,22,23]. Considering these shortcomings, interest in non-pharmacological treatments that can be combined with drug treatment has increased. There is much evidence for the effectiveness of non-pharmacological interventions on depression. Non-pharmacological intervention such as cognitive behavioral therapy (CBT) [24,25], exercise [26], and yoga [27] are widely used to treat depression as non-pharmacological interventions. For example, a meta-analysis of 115 studies found that CBT is an effective treatment strategy for depression, and that it is much more effective to combine it with pharmacotherapy than to treat it with pharmacotherapy alone [28].

Forest therapy can be used as one of these non-pharmacological treatments. Forest therapy is an activity that utilizes the forest environment to improve human health. The methods applied in forest therapy vary considerably. The core element of forest therapy is to recognize the environment of the forest, focusing on the five senses, and meditation, walking, and various types of play can be performed [29]. A number of previous studies have reported the positive benefits of forest therapy on mental health. For example, forest therapy has been reported to reduce psychological stress or mental fatigue [30], anxiety [31,32], and improve mood [33,34], as well as improve quality of life [35,36].

In addition, in recent years, there has been an increasing number of studies showing the possibility that forest therapy can improve depression and that it can be used as an alternative therapy instead of traditional treatments [37,38,39,40]. For example, Kotera et al. [37] conducted a meta-analysis on six RCT studies and showed an effect size of −2.54 (95% CI: 3.56–1.52, *p* < 0.05) for depressive symptoms in forest therapy. A meta-analysis of the effects of forest therapy on depression in 13 studies by Rosa et al. [38] also showed an effect size of 1.18 (95% CI: 0.86–1.50, *p* < 0.001), and participants in the forest therapy group were 17 times more likely to achieve remission and 3 times more likely to decrease 50% of depression symptoms than the no intervention/usual care group. Yeon et al. [39] reviewed 18 studies and reported that forest therapy has a large effect size on depression improvement (Hedges’s g = 1.133; 95% CI: −1.491–−0.775, *p* < 0.001). Kang et al. [40] showed an effect size of 1.36 (95% CI = 0.55–2.17, *p* < 0.001) as a result of a meta-analysis of five studies using depression-related indicators. In other words, it was shown that forest therapy is an effective intervention in improving depression through systematic literature review and meta-analysis.

However, although many previous studies have reported that forest therapy is effective in improving depression, studies exploring the direct link between forest therapy and depression are insufficient. Previous research was conducted on the general public or other patients with diseases such as university students [41], elementary students [42], full-time employees [36], patients with chronic stroke [43], and cancer patients [44]. Furthermore, most of the previous studies were conducted without clinical experts such as psychiatrists, so clinical judgment on the forest therapy effect on depression patients was impossible. Previous studies used indicators evaluating the variables of the depression area among detailed items such as the Profile of Mood States [31,34,45], Stress Response Inventory [46], Multiple Mood State [47], and the Depression Anxiety and Stress Scale [48], rather than objectified evaluation indicators to determine symptoms of depression patients. In addition, most of the studies reported so far have been performed in dense forests far from the city [30,31,32,33,34,35,36]. Such forests have beautiful natural scenery but lack accessibility, so they can not only be limited to use by depression patients but also have sustainability problems.

Therefore, this study aimed to investigate the effects of a forest therapy program using urban forests on depression disorders. In addition, we investigated sleep quality and somatization symptoms closely related to depression in patients with depression.

## 2. Materials and Methods

### 2.1. Participants

The software G*Power 3.1 (University of Düsseldorf, Düsseldorf, Germany) was used to decide the appropriate sample size. It was found that the total sample size was required to be 52. Based on these values, the total sample size was adjusted to 55. The effect size was set to 0.798, with a significance level of 0.05, and the power value was set to 0.8.

Accordingly, fifty-five depression patients (mean age, 37.31 ± 10.27 years) were recruited for the field experiment. The participants were 8 males and 47 females. This initial survey ensured that participants were willing to participate in the study and confirmed that they met the inclusion and exclusion criteria. The inclusion criteria were (a) being an adult (between ages 20 and 59) and (b) having mild depressive disorder diagnosed by psychiatrists according to DSM-5 and confirmed through the Korean version of Structured Clinical Interview for DSM Disorders (SCID). In contrast, exclusion criteria were (a) having a mental state that makes it impossible to leave the psychiatric ward and (b) movement disorders or other somatic diseases that prevent participation in the study.

Fifty subjects met the inclusion criteria, and five participants dropped out of the study due to personal problems. Of the initial 50 participants, 22 were in the urban forest therapy program group (7 males, 15 females; mean age, 37.82 ± 10.27 years) and 25 in the outpatient control group (1 male, 24 females; mean age, 38.88 ± 10.51 years) completed this study. Three patients discontinued the urban forest therapy program group (one dropped out due to a leg injury, and others were absent more than three times due to personal reasons). No patient dropped out of the control group. Before starting the experiments, we explained the study’s purpose and procedures and obtained their written informed consent. All participants provided informed consent and received USD 200.00 compensation for their participation. The Institutional Review Board of Chungbuk National University (IRB number: CBNU-202203-HR-0041) approved this study.

### 2.2. Study Design

We employed the randomized controlled trial study design. The participants were randomly assigned to either the experimental or control group to secure homogeneity between the urban forest therapy program and the control group. A randomization method was used. A random number generator was used to create a list of random numbers from a minimum value of 1 to a maximum value of 50. Odd numbers were assigned to the experimental group and even numbers to the control group. In total, 25 participants were allocated to the experimental group (received the urban forest therapy program) and the control group (treatment as usual). The experiment was conducted from May to June 2022. The pre-test and post-test were evaluated in the first (May 12 to 13) and last weeks (June 18 to 21) of the study period.

### 2.3. Study Site

The study was conducted at the Seoul forest in Seongdong-gu, Seoul Metropolitan City, South Korea. The Seoul forest is a neighborhood park with sufficient accessibility as it is adjacent to a residential area with a subway station, highway toll gate, and bus stop. The main facilities are five themed parks (Culture and Arts Park, Natural Ecology Forest, Nature Experience Learning Center, Wetland Ecology Center, Han River Waterfront Park), an outdoor stage, a square, an environmental playground, a walkway, an event garden, butterfly greenhouse, and toilets. The size of the forest is about 480,994 square meters. The forest was built in 2005, covered mainly by *Pinus strobus* (DBH is about 22 cm; tree height is about 9–14 m), Ginkgo biloba (DBH is about 20 cm; tree height is about 14–19 m), *Pinus densiflora* (DBH is about 20 cm; tree height is about 8–10 m), Zelkova serrata (DBH is about 20 cm; tree height is about 8–10 m), *Metasequoia glyptostroboides* (DBH is about 44 cm; tree height is about 16–18 m), *Prunus serrulate* (DBH is about 28 cm; tree height is about 5–12 m), and other broad-leaved tree species (*Acer buergerianum*, *Koelreuteria paniculate*, *Ulmus davidiana*, *Quercus mongolica*, etc.). The stand age ranges between 20 and 25 years old.

During the six sessions of the experiment, the weather was pleasant and not raining, with a mean temperature of 21.9 ± 2.9 °C.

### 2.4. Urban Forest Therapy Program

The urban forest therapy program met once a week from 14 May to 21 June 2022, from 10 a.m. to 12 p.m., for a total of six sessions. The urban forest therapy program was developed and distributed according to each appropriate session’s theme based on researchers in forest therapy and forest therapists. The program’s main theme was to alleviate depression symptoms by promoting the mental and physical stability of depression patients using the urban forest. In addition, the purpose was to improve the lifestyle by practicing it in daily life through repeated learning of forest therapy activities.

We constructed a healing module in three stages (Table 1). The first step (“Recognition”) is to explore and clarify emotions. The second step (“Action”) is to stop negative thinking through forest therapy activities. The last step (“Change”) is to break the chain of ruminant thinking and improve lifestyle habits through the daily life of forest therapy. The main activities of this program were “relax the body and mind through stretching”, “walking on the five senses”, “playing emotional card game”, and “breathing and meditation” (Figure 1).

The participants in the control group did not receive any forest therapy activities and received the treatment as usual during the experiment. Treatment as usual included medication and counseling was deemed appropriate. All participants continued to be managed by their general practitioners. During the six-week period, all participants were maintained according to each patient’s previous treatment schedule.

### 2.5. Measurement

The psychological evaluations used the Beck Depression Inventory (BDI), the Hamilton Rating Scale for Depression (HRSD), the Pittsburgh Sleep Quality Index (PSQI), and the Patient Health Questionnaire-15 (PHQ-15) questionnaires. The BDI was developed by Beck [49] and was used for participant screening and depressive status. The BDI is a self-reporting tool designed to evaluate the presence and severity of depression. It consists of 21 items covering the cognitive, emotional, motivational, and physical domains of depression, and each item is scored on a scale of 0 to 3, for a total of 63 points. A score of 0–9 indicates no depression, a score of 10–15 indicates mild depression, a score of 16–23 indicates moderate depression, and a score of 24–63 indicates severe depression. In this study, we used the Korean version of the BDI, which has high reliability [50]. The K-BDI of this study was revealed to have high reliability (Cronbach’s α = 0.94).

The HRSD was developed by Hamilton and Williams [51] and applied by the clinician to measure the level of depression and change in the severity of depression. It consists of 17 structured questions. Nine questions are scored on a 5-point Likert scale (0–4), and eight questions are scored on a 3-point Likert scale (0–2); the total score range is 0–52, and the higher the score, the more severe the depression is. In general, if the total score of this scale is 18 or higher, it is often assumed to indicate depression that requires treatment. In addition, when the total score of this scale is reduced by more than half through treatment, there is a treatment response, and if it is less than 7 points, it is called remission, which is considered that both depressive symptoms and the resulting living disorder have been resolved. In this study, we used the Korean version of the HRSD, which has high reliability [52]. The K-HRSD of this study was revealed to have high reliability (Cronbach’s α = 0.97).

The PSQI was used to evaluate the sleep quality of the participants [53]. This tool is self-reported and consists of seven areas and 18 items. The details are as follows: (1) sleep quality, (2) sleep latency, (3) sleep duration, (4) habitual sleep efficiency, (5) sleep disturbance, (6) use of sleeping medication, and (7) daytime dysfunction. The sleep index can be calculated by adding up the scores of all seven items out of a total of 21 points. The higher the score, the lower the quality of sleep. In this study, we used the Korean version of the PSQI, which has high reliability [54]. The K-PSQI of this study was revealed to have high reliability (Cronbach’s α = 0.78).

The Patient Health Questionnaire-15 (PHQ-15), a self-administered test developed by Kroenke et al. [55] to diagnose psychiatric illnesses in a primary care setting, was used to assess somatic symptoms. The PHQ-15 comprises 15 items, and each item is rated on a 3-point scale (range: 0–2). The total score reflects the severity of the somatic symptoms, where scores ≤ 5 are defined as “low,” 6–10 as “medium,” and ≥11 as “high.” In this study, we used the Korean version of the PHQ-15, which has high reliability [56]. The K-PHQ-15 of this study was revealed to have high reliability (Cronbach’s α = 0.86).

### 2.6. Data Analysis

The statistical analyses were performed using SPSS 18.0 Windows (SPSS, Chicago, IL, USA). Descriptive statistics comprised means, standard error, and percentage to present outcome variables. Paired *t*-tests were used to compare participants’ psychological states between pre- and post-tests for each group (urban forest therapy program and control group). In addition, analysis of covariance (ANCOVA) was used to compare the effects between groups before and after the intervention. When significant differences appeared in covariance analysis, the Bonferroni test was performed as a post-hoc test for comparison between groups. All statistical tests used a *p*-value of <0.05 as the significance level.

## 3. Results

### 3.1. Beck Depression Inventory (BDI)

The results of the paired *t*-test between pre- and post-test BDI scores for each group are presented in Table 2. As shown in Table 2, there was a significant decrease in BDI scores for the urban forest therapy program group after six sessions of the urban forest therapy activities (t = 2.605, *p* = 0.017). On the other hand, there were no significant changes in the control group participants’ BDI scores (t = 0.942, *p* = 0.356).

In addition, from the results of analysis using ANCOVA to find out the difference in BDI score changes between the two groups (Table 3), the urban forest therapy program group had a significantly lower BDI score than the control group (F = 4.830, *p* = 0.033).

### 3.2. Hamilton Rating Scale for Depression (HRSD)

The results of the paired *t*-test between pre- and post-test HRSD scores for each group are presented in Table 4. As shown in Table 4, there was a significant decrease in HRSD scores for the urban forest therapy program group after six sessions of the urban forest therapy activities (t = 10.035, *p* < 0.001). Moreover, there was a significant decrease in the control group participants’ HRSD scores (t = 4.643, *p* < 0.001).

On the other hand, from the results of analysis using ANCOVA to find out the difference in HRSD score changes between the two groups (Table 5), the urban forest therapy program group had a significantly lower HRSD score than the control group (F = 42.899, *p* < 0.001).

### 3.3. Pittsburgh Sleep Quality Index (PSQI)

The results of the paired *t*-test between pre- and post-test PSQI scores for each group are presented in Table 6. As shown in Table 6, there was a significant decrease in PSQI scores for the urban forest therapy program group after six sessions of the urban forest therapy activities (t = 4.395, *p* < 0.001). On the other hand, there were no significant changes in the control group participants’ PSQI scores (t = 0.413, *p* = 0.683).

In addition, the results of analysis using ANCOVA to find out the difference in PSQI score changes between the two groups (Table 7), the urban forest therapy program group had a significantly lower PSQI score than the control group (F = 10.377, *p* = 0.002).

### 3.4. Patient Health Questionnaire-15 (PHQ-15)

The results of the paired *t*-test between pre- and post-test PHQ-15 scores for each group are presented in Table 8. As shown in Table 8, there was a significant decrease in PHQ-15 scores for the urban forest therapy program group after six sessions of the urban forest therapy activities (t = 3.919, *p* = 0.001). On the other hand, there were no significant changes in the control group participants’ PHQ-15 scores (t = 0.585, *p* = 0.564).

In addition, the results of analysis using ANCOVA to find out the difference in PHQ-15 score changes between the two groups (Table 9), the urban forest therapy program group had a significantly lower PHQ-15 score than the control group (F = 6.836, *p* = 0.012).

## 4. Discussion

This study evaluated the effects of forest therapy programs using urban forests on depression symptoms, sleep quality, and somatization symptoms of depression patients. This study showed that urban forest therapy programs not only relieve depression symptoms in depression patients but also improve sleep quality and somatization symptoms.

So far, many empirical studies have shown that forest therapy programs provide psychological health benefits for participants [57,58,59]. However, research on the effectiveness of forest therapy programs for mental disorders such as depression is insufficient. In addition, few studies have been reported based on urban forests accessible to depression patients. Therefore, this study suggests the possibility of using a forest therapy program based on urban forests to alleviate depression disorder.

In this study, the urban forest therapy program reduced the BDI scores of depression patients. This is consistent with previous studies such as chronic stroke patients [43], cancer patients [44], and alcoholic patients [60]. The BDI score in the urban forest therapy program group averaged 17.95 ± 2.78 points before the intervention, which indicated “moderate depression”. After implementing the six sessions, the BDI score in the urban forest therapy program group averaged 11.68 ± 2.24 points, which indicated that depression was alleviated to “mild depression.” On the other hand, the control group did not change to “moderate depression” in the pre-test (20.56 ± 2.32 points) and the post-test (19.16 ± 2.69 points). In addition, these findings show that the urban forest therapy program can achieve similar therapeutic effects compared to cognitive behavioral therapy (CBT), even though it was conducted for a relatively short period of six sessions. The results of this study were a reduction in depressive symptoms of about 34% compared with baseline as measured by the BDI. For example, Scott et al. [61] conducted a randomized controlled trial of CBT and treatment as usual (TAU) for 58 weeks in 48 patients with major depressive disorders. After seven weeks, participants in the CBT group decreased the symptoms of depression by 39% compared to the baseline measured by BDI, and the TAU group decreased by 24%. Serfaty et al. [62] conducted a randomized controlled trial of CBT, TAU plus talking control (TC), and TAU to determine the clinical efficacy of CBT over four months in 204 elderly patients with depression. As a result, the CBT group decreased the symptoms of depression by 33%, the TC group decreased by 23%, and the general treatment group decreased by 27%. In addition, in a study by Power et al. [63], a randomized controlled trial of CBT, interpersonal psychotherapy (IPT), and TAU were conducted in 71 patients with depression. After five months, the CBT group showed a 39% decrease in depressive symptoms compared to the baseline measured by BDI, a 54% decrease in the IPT group, and a 24% decrease in the TAU group. However, this study cannot be generalized because the severity of the patient’s depression symptoms and the socio-demographic characteristics are different compared to previous studies on cognitive behavioral therapy. Therefore, future studies need to compare the effects of forest therapy programs and other non-pharmacological interventions.

In this study, the level of depression in the urban forest therapy program group decreased compared to the control group for HRSD, a measure evaluated by clinicians. The findings are consistent with previous studies showing that forest therapy programs can alleviate depression [43,64]. However, in the control group, the HRSD score decreased six weeks after the pre-test. These results are likely because participants in the control group received the usual care during the study period according to the treatment schedule.

The results of this study are also significant in that in the case of the urban forest therapy program group, the HRSD score reaches below the HRSD standard of seven points after the implementation of the six-session urban forest therapy program. This is called remission, which promotes the return of daily and social activities. Before conducting the study, the HRSD score of participants in the urban forest therapy program was 30.27 ± 2.35 points, and that of the control group was 29.28 ± 3.13 points, showing depression. However, in the case of the experimental group that received the six-session urban forest therapy program, it decreased significantly to 6.41 ± 0.95 points. It improved significantly from “depression” to “remission”. In addition, it can be interpreted that the total score of the scale has been reduced by more than half, resulting in a treatment response. On the other hand, in the case of the control group, daily life was requested to be performed in addition to outpatient treatment for 6 weeks. As a result, after 6 weeks, the HRSD score improved from 29.28 ± 3.13 to 21.21 ± 2.93 points, but there was no treatment reaction.

In this study, the experimental group performed usual care and urban forest therapy programs simultaneously, and the control group received only usual care without intervention. Therefore, the results of this study indicate that it is more effective to alleviate depression symptoms when patients with depression perform usual care and forest therapy together than only receiving usual care. Furthermore, this measure can be seen as a significant result in that patients showed such improvement in the objective indicators evaluated by experts with medical knowledge, not subjectively reporting the level of depression.

The results of this study show that the urban forest therapy program improved the quality of sleep for depression patients. The findings are consistent with some previous studies showing that nature-based interventions help relieve insomnia [65,66]. For example, Morita et al. [65] revealed that when walking in forest areas, sleep time and sleep quality improved. Kim et al. [66] studied the effect of a five-night, six-day forest therapy program on the resolution of insomnia in women in menopause. As a result of the study, participants reported that sleep efficiency increased and waking after sleep onset decreased after implementation before the forest therapy program was implemented. Shen et al. [67] reported a significant improvement in sleep quality as a result of conducting a six-session horticultural program for 48 elderly people aged 70 to 93.

Patients with depression with sleep disorders may experience more severe symptoms and difficulty in treatment [68]. Persistent insomnia is also known as the most common residual symptom in patients with depression and can affect poor clinical outcomes as it is considered an important predictor of depression recurrence [68,69]. In other words, improving the quality of sleep improves the outcome of depression [70,71]. Therefore, the results of this study are believed to have helped improve depression by improving the quality of sleep.

In addition, the results of this study showed that urban forest therapy programs relieve somatization symptoms in depressive disorder. The average PHQ-15 score in the urban forest therapy program group was 11.05 ± 1.33 points before the intervention, which was “high”. After the implementation of the urban forest therapy program in the sixth session, it was 8.41 ± 1.36 points, which was alleviated to a “medium”. On the other hand, the control group did not change to “high” in the pre-test (11.58 ± 1.02 points) and the post-test (11.25 ± 1.24 points). The PHQ-15 is a self-reported measure that has been used to evaluate somatization symptoms in psychiatric patients in several previous studies [72,73]. To the best of our knowledge, it has not been used to study the effects of nature-based interventions, including forest therapy, on the somatization symptoms of depression patients and general public. In this study, it is significant that the urban forest therapy program using PHQ-15 alleviated the somatization symptoms of depression patients.

Our findings suggest that healthy five-sense stimulation in forest environments, such as forest landscapes, forest sounds, scents, and various tactile factors, may be associated with improving sleep quality and alleviating somatization symptoms as well as depression. The main activity of the urban forest therapy program was to focus on the five senses of sight, smell, hearing, touch, and taste by utilizing the urban forest environmental factors for each session. Many previous studies reported that five-sense stimulation such as forest landscape [74,75], forest-induced auditory stimulation [76,77], scent [78], and touch [79,80] improve mental and physical relaxation and stress recovery.

In addition, regular physical activities such as stretching, playing, and walking through the urban forest therapy program are thought to have a positive effect on depression patients. The World Health Organization [81] and the U.K. National Institute for Health and Clinical Excellence (NICE) guidelines recommend regular physical exercise as a standard complementary treatment for depression [82]. The beneficial effects of physical exercise in the treatment of depression have previously been demonstrated through several meta-analyses [26,83,84].

Another important factor is that the urban forest therapy program encouraged depression patients to be exposed to sufficient sunlight. Combined stimulation of natural sunlight affects sleep quality and depression [85,86]. Some studies also reported that low vitamin D levels in the blood were associated with symptoms of depression [87,88,89]. Because vitamin D is known as the sunshine vitamin [90], exposure to sunlight is known to help prevent or improve depression. This program was conducted from 10 a.m. to 12 a.m., and it is thought that the depression symptoms of patients improved because it led them to be naturally exposed to sunlight through various forest activities such as stretching, walking, and emotional cards. For example, Anouti et al. [91] conducted an online survey of 245 participants evaluating depression, physical symptoms, and levels of day and night activity. Multivariate logistic regression analysis showed that only daytime outdoor activities, not night outdoor activities, were associated with a clinically significant reduction in the risk of depression and physical symptoms. This may be due to an increase in natural sunlight exposure as one of the factors for improving depression through the forest therapy program, but further studies are needed to confirm this conjecture. In future studies, it is necessary to study the relationship between forest therapy programs with sunlight exposure, vitamin D synthesis, and improvement of depression symptoms.

In addition, the study was conducted using urban forests with excellent access to the city, rather than dense forests far from the city. Urban green spaces, such as urban forests, can provide cost-effective, simple, and accessible ways to prevent depression and improve the quality of life and health of depressed patients. For example, Zhou et al. [92] reported that urban green space ratios were negatively associated with depression prevalence among urban middle-aged (OR = 0.79, *p* < 0.05) and elderly residents (OR = 0.75, *p* < 0.05), and public recreational green space helped reduce depression in the elderly (OR = 0.77, *p* < 0.05). In addition, in a study by Min et al. [93], adults with many parks around their residences had a 16–27% lower risk of depression and suicide than adults with few parks around their residences. In another study by Mukhejee et al. [94], people living near the park were 3.1 times less likely to develop depression (95% CI: 1.4–7.0) than people living far away from the park. Accordingly, it seems that the existence of urban green spaces around the residence can positively affect the prevention and treatment of depression.

Therefore, it is important to prepare a way for depression patients to improve their daily life and make it easy to access by utilizing the recovery environment such as urban forests and urban green spaces. To do so, it is important to provide sufficient and accessible urban green space as a whole. In addition, the government needs to consider establishing appropriate healthcare policies to encourage those with depression disorder to interact more with public urban green spaces.

This study has several limitations. First, the sample size of this study was small. When the sample size was calculated for the effect size, 52 participants were needed, but only 47 participants were analyzed due to dropout. In order to generalize the results of the study, it is necessary to use a group with different socio-demographic characteristics with a large sample size matched to calculate the effect size in future studies. Second, this study did not determine whether patients were in contact with nature other than through the urban forest therapy program. In future studies, it is necessary to investigate the frequency and time of natural exposure in addition to the urban forest therapy program. Third, this study showed a difference in gender ratio in the randomized allocation process. There was no stratification involved in the randomization procedure, but in retrospect, it would have been useful to stratify gender to equalize the number of men and women. Fourth, it was not possible to confirm the drugs taken by the patients who participated in this study. In future studies, it is necessary to analyze the effect using the dosage as a covariate. Fifth, the long-term effects of forest therapy were not investigated, and the mechanisms underlying the beneficial effects of forest therapy were not identified. For depression patients to return to their daily lives, they had to continue remission for six months, but this study did not identify the condition after six months. Sixth, this study did not identify the healing factors of forest therapy and the effects of seasonal factors such as atmosphere and weather on participants’ depressive symptoms. These may be potential factors that can affect the effectiveness of forest therapy, which needs to be revealed in future studies. Seventh, in this study, the control group performed treatment as usual. Some control groups can use forests for leisure, and this experience may affect the results of this study. Therefore, in future studies, it is necessary to set up participants who spend time in the forest without giving instructions while performing treatment as usual as a control group.

Despite these limitations, the results of this study support evidence that forest therapy, which is part of nature-based intervention, can be used as a non-pharmacological intervention to improve depression. In addition, urban forests offer remarkable strengths, being easily available to depression patients as well as healthy people at any time.

## 5. Conclusions

This study showed that the urban forest therapy program significantly improves sleep quality and somatization symptoms as well as depression symptoms in depression patients. These findings demonstrate that the urban forest therapy program could be presented as a potential strategy to improve the mental health of depression patients. Therefore, forest therapy could be used as a non-pharmacological intervention to improve depression.

## Figures and Tables

**Figure 1 ijerph-20-00507-f001:**
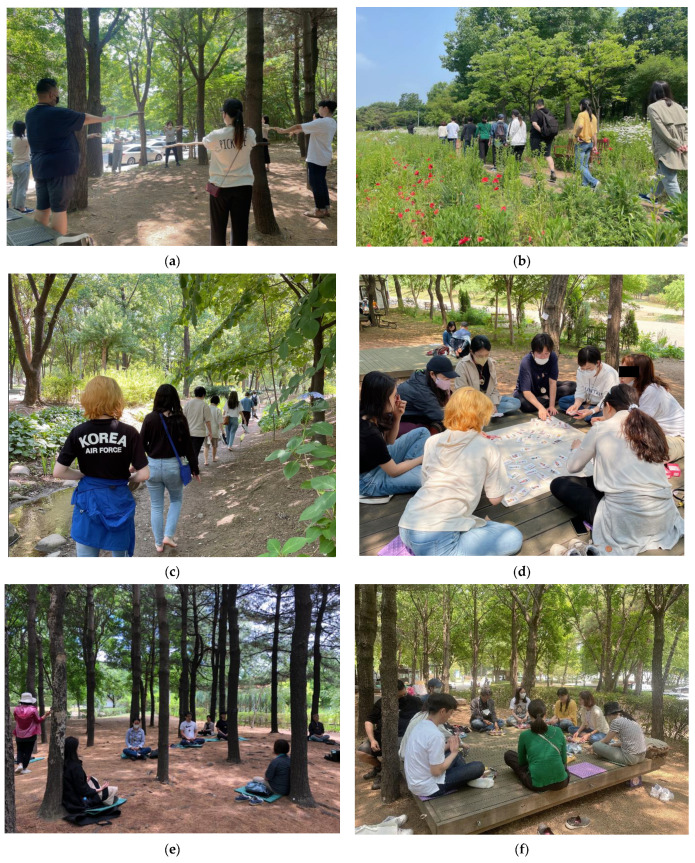
Urban forest therapy program. (**a**) Stretching; (**b**) walking on the five senses (vision); (**c**) walking on the five senses (touch); (**d**) playing an emotional card game; (**e**) breathing and meditation; (**f**) sharing and finishing.

**Table 1 ijerph-20-00507-t001:** Activity details of the urban forest therapy program.

Healing Module	Recognition (1, 2 Sessions)	Action (3, 4 Sessions)	Change (5, 6 Sessions)
Objective	Exploring and clarifying my feelings	Experience stopping negative thoughts through forest activities	Leads to break the loop of rumination thinking Daily life of forest activities
Program Activities	Stretching Walking on the five senses (Visual, Sound) Playing emotional card game (To know about me) Walking in meditation	Stretching Walking on the five senses (Smell, Taste) Playing an emotional card game (Match the other person’s emotions) Breathing meditation	Stretching Walking on the five senses (Touch, All senses) Playing an emotional card game (Break the thought chain) Breathing meditation

**Table 2 ijerph-20-00507-t002:** The result of paired *t*-test analysis of Beck Depression Inventory.

Variable	UFTP (*n* = 22)				Cont. (*n* = 25)			
	Pre-Test	Post-Test	t	*p*	Pre-Test	Post-Test	t	*p*
	M (SE)	M (SE)			M (SE)	M (SE)		
BDI	17.95 (2.78)	11.68 (2.24)	2.605	0.017	20.56 (2.32)	19.16 (2.69)	0.942	0.356

Note: UFTP: urban forest therapy program group; Cont.: control group.

**Table 3 ijerph-20-00507-t003:** The results of the analysis of covariance of Beck Depression Inventory.

Variable	Sum of Squares	df	Mean Square	F	*p*
BDI					
Pre-test	3284.283	1	3284.283	42.756	<0.001
Group	371.047	1	371.047	4.830	0.033
Error	3379.850	44	76.815		

**Table 4 ijerph-20-00507-t004:** The result of paired *t*-test analysis of Hamilton Rating Scale for Depression.

Variable	UFTP (*n* = 22)				Cont. (*n* = 25)			
	Pre-Test	Post-Test	t	*p*	Pre-Test	Post-Test	t	*p*
	M (SE)	M (SE)			M (SE)	M (SE)		
HRSD	30.27 (2.35)	6.41 (0.95)	10.035	<0.001	29.28 (3.13)	21.24 (2.93)	4.643	<0.001

Note: UFTP: urban forest therapy program group; Cont.: control group.

**Table 5 ijerph-20-00507-t005:** The results of the analysis of covariance of Hamilton Rating Scale for Depression.

Variable	Sum of Squares	df	Mean Square	F	*p*
HRSD					
Pre-test	2719.151	1	2719.151	42.117	<0.001
Group	2769.661	1	2769.661	42.899	<0.001
Error	2840.727	44	64.562		

**Table 6 ijerph-20-00507-t006:** The result of paired *t*-test analysis of Pittsburgh Sleep Quality Index.

Variable	UFTP (*n* = 22)				Cont. (*n* = 25)			
	Pre-Test	Post-Test	t	*p*	Pre-Test	Post-Test	t	*p*
	M (SE)	M (SE)			M (SE)	M (SE)		
PSQI	20.50 (1.88)	15.05 (1.85)	4.395	<0.001	19.64 (1.59)	19.24 (1.59)	0.413	0.683

Note: UFTP: urban forest therapy program group; Cont.: control group.

**Table 7 ijerph-20-00507-t007:** The results of the analysis of covariance of Pittsburgh Sleep Quality Index.

Variable	Sum of Squares	df	Mean Square	F	*p*
PSQI					
Pre-test	1898.941	1	1898.941	70.179	<0.001
Group	280.773	1	280.773	10.377	0.002
Error	1190.574	44	27.058		

**Table 8 ijerph-20-00507-t008:** The result of paired *t*-test analysis of Patient Health Questionnaire-15.

Variable	UFTP (*n* = 22)				Cont. (*n* = 25)			
	Pre-Test	Post-Test	t	*p*	Pre-Test	Post-Test	t	*p*
	M (SE)	M (SE)			M (SE)	M (SE)		
PHQ-15	11.05 (1.33)	8.41 (1.36)	3.919	0.001	11.58 (1.02)	11.25 (1.24)	0.585	0.564

Note: UFTP: urban forest therapy program group; Cont.: control group.

**Table 9 ijerph-20-00507-t009:** The results of the analysis of covariance of Patient Health Questionnaire-15.

Variable	Sum of Squares	df	Mean Square	F	*p*
PHQ-15					
Pre-test	1314.581	1	1314.581	145.975	<0.001
Group	61.566	1	61.566	6.836	0.012
Error	387.237	44	9.006		

## Data Availability

Not applicable.

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
