# Peer review of "Effects of Urban Forest Therapy Program on Depression Patients"

_ijerph, 2022, doi:10.3390/ijerph20010507_

Round 1

Reviewer 1 Report

This study aimed to examine the effects of the urban forest therapy program on depression symptoms, sleep quality, and somatization symptoms of depression patients. A randomized controlled trial (RCT) design was employed. A total of 47 depression patients participated in this study. Results of this study revealed that depression patients in the urban forest therapy program had significantly alleviated depression symptoms, and improved sleep quality and somatization symptoms compared to the control group.

The revised article evaluates an interesting aspect of current therapies such as using the beneficial effects of nature on emotions. There is a good design to answer the aim of the paper. However, there are some aspects that need to be improved before publication.

Firstly, the sample is small, which should be considered a limitation. In addition, it has not been described whether these patients have more contact with nature in addition to the sessions described. Therefore, both aspects should be considered as limitations to be worked on in future studies.

On the other hand, I do not understand why use two depression questionnaires. The authors should consider keeping only one questionnaire, since if both measure depression it is a redundant result.

Another important aspect to improve is to describe the routine activities of the control group. The main treatment is described, but it is not known what the control group does. It is necessary to know this in order to assess the real effect of the treatment.

Finally, the discussion is correct and oriented to compare with other studies using similar therapies, but I consider that a critical aspect to discuss is the value that these new therapies bring with respect to classical therapies such as Cognitive Behavioural Therapy CBT). In fact, I believe that the results obtained in this study should be compared numerically with other studies using classical therapies. How much score reduction can a psychotherapy treatment reduce in a similar time? I think this is the place where this type of study should go in order to show whether the effects found are minor, similar or superior to classical therapies. In this sense, I think the introduction would also benefit from information regarding the effectiveness of more classical psychotherapy treatments.

minor points:

·        Table 3 is cut off

·        In lines 53-54 it talks about medication and biological treatments, what is the difference? Clarify this information.

·        lines 105-107: the 2nd inclusion criterion is not clear.

Author Response

Dear Editor and reviewer,

We would like to express our sincere gratitude for your kind consideration and comments on our manuscript. According to reviewers’ comments and suggestions, we revised the manuscript as follows:

(We marked the revision to the reviewer's comment in red)

  1. We added to the limitation that the sample size is small and that our study does not reveal whether patients have more contact with nature (line 443-450).
  2. Thank you again for your sincere coment. We used two depression evaluation tools at the same time because we thought that using a self-report measurement tool (BDI) and a measurement tool evaluated by clinicians (HDRS) could more objectively evalluate the effect of improving depression symtoms in the urban forest therapy program.
  3. The control group did not receive a forest therapy program during the experiment period, but received the treatment as usual. These limitations are described in this article (line 188-192, 463-466).
  4. We added information on the effectiveness of classical psychotherapy to the introduction. In addition, we added content the our results in this study compared numerically with other studies using classical therapies in discussion (line 65-72, 313-333).
  5. We modified Table 3 (line 256).
  6. We described the terms medication and biological treatment in duplicate. So we unified the term with medication (line 60)
  7. We modified 2nd inclusion criterion (line 125-127).

Reviewer 2 Report

This study evaluated the effect of urban forest therapy on depression symptoms. Study shows improvements in the many aspects of wellbeing of forest therapy participants. However, the major limitation of the study is that control group did not follow similar program. Therefore it is impossible to distinguish what part of the improvements are due to forest visits and what part is due to other reasons such as meditation, physical activity, being part of the group or other reason related to the program.

Abstract:

Explain in your abstract the background of the study. Why to study forest therapy and depression?

Please explain already in the abstract what is urban forest therapy in practice.

Introduction:

Depression affects 3.5% of the population… is the population here the whole world? Add citation.

Somatization: please explain what that means?

row 44-45: general pain and pain… please rephrase.

incidental treatment: please explain what that means?

RCT studies, please open RCT?

MD, please open?

Rows 73-74: 17 times more than who?

Rows 81-91: please add references to the studies with the limitations you describe. It was rather hard to really get what is the limitations your study is addressing and how?

Methods:

Your original analysis suggested that 55 patients were needed for the study. However, you end up having 22+25 = 47 patients. How does this influence on your study?

What is the sex ratio in groups after exclusion?

Study is well described. Maybe you could include a map showing the areas which were used in the study?

Different scales to measure depression etc. are well-described. It is also great that Korean versions exists and are used.

Add paragraph explaining what background information was collected from participants?

Wouldn’t it be important to adjust your analyses for sex, socioeconomic status, age and potentially other variables? Your data can also be too small for this..

When the pretest and posttest measurements were done?

Table 3 is partly hidden.

Row 253: increase, do you mean decrease?

Add table explaining the background features of the both groups.

Discussion:

Table 4 shows that both groups had decreased scores. Why?

Author Response

Dear Editor and reviewer,

We would like to express our sincere gratitude for your kind consideration and comments on our manuscript. According to reviewers’ comments and suggestions, we revised the manuscript as follows:

(We marked the revision to the reviewer's comment in blue)

  1. We added to the Abstract why forest therapy, depression, and urban forest therapy should be studied as the background of the study (line 15-19).
  2. We added reference to the depression population as it means the world (line 37-38).
  3. We added a description of somatization (line 47-49).
  4. We modified terms for General pain and pain (line 51-52).
  5. We modified the “incidental treatment” to “alternative treatment” to deliver the meaning of the term, and added an explanation (line 83).
  6. We added in the introduction that it was 17 times higher than the control group (line 89).
  7. We added references to studies with limitations described (line 98-100, 103-108).
  8. We needed 52 depression patients for this study. Therefore, 55 depression patients were recruited in consideration of dropouts. However, more participants were eliminated, allowing them to analyze 47 depression patients. These limitations have been added to the discussion (line 443-450).
  9. We added the sex ratio of the group after exclusion (line 131-133, 450-453).
  10. We added a map showing the area used un the study (line 172).
  11. We were not able to collect data on socioeconomic demographic characteristic other than age and gender for participants. We think this is a limitation of our research. We will take note of your comments and supplement them in future studies.
  12. We conducted pretest and posttest in the first and last weeks of the study period. We added this content (line 150-151).
  13. We modified Table 3 (line 256).
  14. We made a mistake. It is correct that there has been a decrease. We modified the content (line 273, 279).
  15. We added the reason for the decrease in HRSD scores in both groups (line 337-340, 354-357).

Round 2

Reviewer 1 Report

Dear authors,

Thank you for changing the manuscript following the reviewer suggestions. I feel that the manuscript has improved and can be more useful to the readership.

Author Response

Thank you very much for your suggestions that allow us to improve our manuscript.

Reviewer 2 Report

Thank you for responding my concerns. I would still like to see the limitation regarding the role of forest itself, i.e., control group did not follow any program, to be discussed. Further, the map you added is inadequate and can be either removed or replaced with more detailed one showing the specific spots or routes the group used.

Author Response

Dear Editor and reviewer,

We would like to express our sincere gratitude for your kind consideration and comments on our manuscript. According to reviewers’ comments and suggestions, we revised the manuscript as follows:

(We marked the revision to the reviewer's comment in blue)

  1. We added to the limitation that the control group did not follow any program (line 463-466).
  2. Thank you very much for your suggestion. We removed insufficient maps.